# Medicinal Properties of *Anchusa strigosa* and Its Active Compounds

**DOI:** 10.3390/molecules27238239

**Published:** 2022-11-25

**Authors:** Ludmila Yarmolinsky, Arie Budovsky, Boris Khalfin, Leonid Yarmolinsky, Shimon Ben-Shabat

**Affiliations:** 1Eastern R&D Center, Kiryat Arba 9010000, Israel; 2Research & Development Authority, Barzilai University Medical Center, Ashkelon 7830604, Israel; 3Faculty of Health Sciences, Ben-Gurion University of the Negev, Beer-Sheva 8410501, Israel; 4Arnie Miller Laboratories, Beer-Sheva 8430713, Israel

**Keywords:** *Anchusa strigosa*, wound healing, anti-microbial compounds

## Abstract

*Anchusa strigosa* is a widespread weed in Greece, Syria, Turkey, Lebanon, Israel, Jordan, and Iran. The purpose of this study was to identify the phytochemicals of *Anchusa strigose* and estimate the pro-wound healing (pro-WH) and antimicrobial activities of its active compounds. An identification of volatile compounds was performed by GC/MS analysis; HPLC, LC-ESI-MS, and MALDI-TOF-MS were also applied. Our results demonstrate that two specific combinations of compounds from *A. strigosa* extract significantly enhanced WH (*p* < 0.001). Several flavonoids of the plant extract, including quercetin 3-*O*-rutinoside, kaempferol, kaempferol 3-*O*-β-rhamnopyranosyl(1→6)-β-glucopyranoside, and kaempferol 3-*O*-α-rhamnopyranosyl(1→6)-β-galactopyranoside, were effective against drug-resistant microorganisms. In addition, all the above-mentioned compounds had antibiofilm activity against *Escherichia coli* and *Salmonella enteritidis*.

## 1. Introduction

*Anchusa strigosa* (strigose bugloss or prickly alkanet) is a widespread weed, with a rosette of leaves at its base and an inflorescence stem reaching one meter or more [1]. This plant belongs to the *Boraginaceae* family, and it is native to Greece, Syria, Turkey, Lebanon, Israel, Jordan, and Iran. The specimens used in this research were collected in the Judea region (Israel). This region has a peculiar geography that influences local herbal populations [1]. The plants from the Judea region are exposed to permanent stress, which stimulates production of high quantities of various phytochemicals with promising therapeutic properties [2,3,4,5,6,7,8,9,10].

Although many publications are devoted to the plants of this region, to the best of our knowledge, *A. strigosa* medicinal properties have not been reported. In general, some studies demonstrated medicinal properties of *A*. *strigosa* from other regions including anti-ulcer [11], pro-wound healing (WH) [12] and antioxidant activities [13,14]. Yet, it is still unknown which compounds are responsible for their therapeutic effects.

In this publication, we focus on investigating pro-WH properties since WH is one of the most important public health problems worldwide. For example, chronic wounds annually affect 6.5 million patients in the United States [15]. Indeed, the number of chronic wounds has a tendency to increase due to the rise in age-related conditions and pathologies, such as diabetes, obesity, and cardiovascular diseases [15]. The calling card of WH is inadequate efficacy and a number of serious adverse effects associated with the widespread use of common therapeutic agents. For example, glucocorticoids are effective in many cases, but they have many side effects, including the promotion of wound infections [15]. In addition, deviations from regular WH could lead to diverse pathological conditions, from slow or ineffective repair (e.g., diabetic, pressure and ischemic skin ulcers, mainly in older people) to fibrosis (e.g., hypertrophic scars, keloids, scleroderma, mainly in young and middle-aged people [16,17].

As suggested by us and others [10,13,18,19], another factor that retards the process of WH are infections, mainly of the bacterial origin.

With regard to bacterial infections, the combined influence of environmental factors [20,21] and overuse of antibiotics [21] can cause the widespread appearance of antibiotic-resistant microorganisms. Some modern approaches against antibiotic-resistant microorganisms include a reduction in antibiotic consumption, preservation of existing therapeutics, and development of new antibiotics [21]. Unfortunately, the above-mentioned measures have not been effective so far.

Gram-negative bacteria are more widespread in hospital patients than Gram-positive bacteria [22]. *Escherichia coli*, *Klebsiella pneumoniae*, *Acinetobacter baumannii*, *Serratia marcescens* and *Salmonella enteritidis* are Gram-negative species that are responsible for major health problems worldwide.

In addition, many microorganisms become more dangerous because of quorum sensing (QS) [23]. QS is one of the major signaling mechanisms that directly contributes to biofilm formation [24]. Biofilm formation is one of the challenging problems in treating infections as bacteria in biofilms are extremely resistant to the antibacterial agents.

Thus, there is an urgent demand for novel anti-microbial compounds that must be effective, non-toxic and able to suppress QS. Medicinal plants hold enormous potential in this context as *A. strigosa* is a rich reservoir of such compounds.

In summary, this study aimed to identify active compounds of *A. strigosa* with pro-WH and antimicrobial activities.

## 2. Results

### 2.1. Identification of Active Compounds

The following compounds of *A. strigosa* were identified (Table 1). Bioassay-guided fractionation of the extract allowed us to determine that only one fraction eluted with 80%-MeOH had pro-wound healing and anti-microbial properties. Different analytical methods used in this study confirmed each other, and the comprehensive use of various analytical methods was very helpful for increasing the probability of compound identification. The Appendix A includes the GC-MS and LC-ESI-MS chromatograms of extracts of *Anchusa strigosa*.

### 2.2. Pro-Wound Healing (WH) Properties

Taking into consideration an important role of dermal fibroblasts in skin WH, we used HDF (human dermal fibroblasts) cell lines in the current study. The crude extract was not toxic at a concentration below 900 µg/mL for HDF cells; quercetin 3-*O*-rutinoside and ellagic acid also had no cytotoxicity at this concentration. The other identified compounds were toxic for the HDF cells at concentrations of more than 100 µg/mL. The ethanol at the used concentrations was not toxic.

The crude extract significantly stimulated pro-WH (*p* < 0.001), when it was added at a concentration of 50 µg/mL (Figure 1). It is interesting to note that, at a concentration of 50 µg/mL, each separate compound did not significantly change the gap closure. Then, the compounds were tested in various combinations. Figure 1 demonstrates that both combinations of the tested compounds, combination 1 and combination 2, had a significant pro-WH effect (*p* < 0.001). Combination 1 of the tested compounds included quercetin 3-*O*-rutinoside at a concentration of 50 µg/mL, ellagic acid at a concentration of 10 µg/mL, and kaempferol 3-*O*-β-rhamnopyranosyl(1→6)-β-glucopyranoside at a concentration of 10 µg/mL. Combination 2 of the tested compounds included kaempferol at a concentration of 10 µg/mL and ellagic acid at a concentration of 1 µg/mL. In addition, both mixtures of compounds were not toxic for cells (unpresented data).

### 2.3. Estimation of Anti-Microbial Properties

The ethanolic diluted extract of *A. strigosa* and its identified compounds (Table 1) were applied for the estimation of their anti-microbial properties in vitro. The concentration of the initial extract was 2.5 mg/mL in PBS, and the diluted concentration was 0.1 mg/mL. The use of a positive control was not possible in this experiment because antibiotics do not influence these strains. The negative control was PBS. The phytochemicals were tested at a concentration of 2 µM. Although every identified compound was examined, only effective compounds are presented in Figure 2. We found that quercetin 3-*O*-rutinoside was less effective than the crude extract, while kaempferol and its glycoside derivatives significantly inhibited all tested microorganisms (*p* < 0.001).

In order to estimate an effect of all identified compounds on biofilm formation, a crystal violet assay was used [25]. Only quercitin 3-*O*-rutinoside, kaempferol, kaempferol 3-*O*-β-rhamnopyranosyl(1→6)-β-glucopyranoside and kaempferol 3-*O*-α-rhamnopyranosyl(1→6)-β-galactopyranoside had antibiofilm activity (Figure 3). The effect of these compounds was significant against *Escherichia coli* and *Salmonella enteritidis* (*p* < 0.001) (Figure 3), while streptomycin (positive control) was less effective against *Salmonella enteritidis* than quercitin 3-*O*-rutinoside, kaempferol, kaempferol 3-*O*-β-rhamnopyranosyl(1→6)-β-glucopyranoside and kaempferol 3-*O*-α-rhamnopyranosyl(1→6)-β-galactopyranoside (Figure 3).

## 3. Discussion

Our previous publications [6,10] showed that plants growing in the Judea region (Israel) have unique chemotypes. For example, when we studied another plant from the same area, *Phlomis viscosa* Poiret, we expected to identify β-Caryophyllene, germacrene D, alloaromadendrene and humulene as bioactive compounds since these compounds were reported to have medicinal activities in plants growing in Turkey [26]. Surprisingly, these compounds were not identified by us, but other compounds were found to possess medicinal properties in the Israeli plants [6,10].

Although MALDI-TOF-MS is more often used for the analysis of polypeptides, the structural heterogeneity of plant polyphenols requires more detailed investigation [27,28]. The different analytical methods used in this study confirmed each other, and the comprehensive use of various analytical methods was very helpful for increasing the probability of compound identification.

The presence of pyrrolizidine alkaloid was confirmed in *Anchusa strigosa* (Table 1) and its presence is also mentioned in the literature [29], but to the best of our knowledge, the rest of the identified compounds (Table 1) were not reported to be present in the investigated weed. We showed that the crude extract was not toxic at concentrations below 900 µg/mL for HDF cells. Its concentration of 50 µg/mL was sufficient for effective pro-wound healing activity (Figure 1), while each identified compound on its own is not active. We succeeded in finding some effective combinations of compounds (Figure 1), proving that the extract has a pro-wound healing activity because of the synergistic interactions between its components. It is interesting that the absence of the desired activity in individual identified compounds of plant extracts is mentioned in the literature [10,30,31]. Further research to identify other compounds of the extract is necessary.

Quercetin 3-*O*-rutinoside, ellagic acid, kaempferol and kaempferol 3-*O*-β-rhamnopyranosyl(1→6)-β-glucopyranoside were identified as the pro-WH components of the extract (Figure 1). Although the pro-WH properties of quercetin 3-*O*-rutinoside, ellagic acid and kaempferol were previously mentioned [15,31,32,33], kaempferol 3-*O*-β-rhamnopyranosyl(1→6)-β-glucopyranoside was not reported as a pro-WH compound. The molecular mechanisms of their actions are unknown. The modes of actions behind the synergistic enhancement of WH by our combinations of compounds are yet to be investigated.

We demonstrated that several flavonoids of *Anchusa strigosa* were effective against drug-resistant microorganisms (Figure 2). The antibacterial activity of kaempferol against drug-resistant *Staphylococcus aureus* was reported [34], but no information is available on the antibacterial activity of kaempferol 3-*O*-β-rhamnopyranosyl(1→6)-β-glucopyranoside and kaempferol 3-*O*-α-rhamnopyranosyl(1→6)-β-galactopyranoside. Although the mode of action of some antibacterial flavonoids was partially revealed [35], the mechanism of action of kaempferol and its glycoside derivatives should be investigated in the future. In addition, all of these compounds demonstrated anti-QS properties (Figure 3).

Altogether, our results indicate the beneficial effect of extracts of *A. strigosa* and its phytochemicals on WH, as well as their anti-microbial properties, including anti-QS abilities and efficacy against drug-resistant bacteria. Thus, there is a need for further and more detailed research on the mechanisms of action of the tested compounds.

## 4. Materials and Methods

### 4.1. Preparation of Plant Material

Leaves and flowers of *A. strigosa* were grounded for gas chromatography/mass spectrometry (GC/MS) analysis. Leaves and flowers of the plant were used for the preparation of the extract as described in our previous publications [10,32,35]. Fractionation of the extract was performed with assistance of reverse phase RP-C18 Sepack column (Supelco, St. Louis, MO, USA); methanol gradients were 0% (*v*/*v*), 20% (*v*/*v*), 40% (*v*/*v*), 60% (*v*/*v*), 80% (*v*/*v*) and 100% (*v*/*v*) [10,36].

### 4.2. Identification of Plant Compounds

An identification of volatile compounds was performed with assistance of GC/MS analysis [6,10,36]. High-performance liquid chromatography (HPLC) (HEKAtech GmbH, Wegberg, Germany), liquid chromatography–electrospray ionization–mass spectrometry (LC-ESI-MS) (Bruker Daltonik GmbH, Bremen, Germany) and Matrix-assisted laser desorption/ionization—time-of-flight—mass spectrometry (MALDI-TOF-MS) (Bruker, Berlin, Germany) were used.

### 4.3. Materials and Bacterial Strains

Lyophilized powders of *Escherichia coli*, *Klebsiella pneumoniae*, *Acinetobacter baumannii*, *Serratia marcescens* and *Salmonella enteritidis* were obtained from ATCC, and drug-resistant bacteria were treated as described in our previous publication [10].

### 4.4. Biofilm Formation Estimation

The crystal violet assay was performed as previously described [37]. The biofilm-producing bacteria were cultured with or without the tested compounds or streptomycin for 48 h at 37 °C. The nonadherent bacteria were removed by washing with sterile PBS, and adherent bacteria were stained for 10 min with a 1% crystal violet solution. A crystal violet assay was performed in 96-well polystyrene plates.

### 4.5. Anti-Bacterial Activity

An equal number of cells were seeded in 96-well plates in 100 μL of appropriate medium with or without the tested compounds. Desired concentration of each chemical was achieved by addition of calculated volume of stock solutions. Plates were incubated at 37 °C and in 5% CO_2_ for up to 5 days. Toxicity was determined either by cell counting once every 24 h or by WST method on the final day, as described by manufacturer [38].

### 4.6. In Vitro Wound Healing Assay

Pro-wound-healing effects were studied in an in vivo, as described in our previous publication [10].

### 4.7. Statistical Analysis

Statistica Version 13.6 for Windows software (StatSoft, Inc., Tulsa, OK, USA) was chosen for statistical data processing. Numbers represent the mean ± standard error from at least three independent experiments, each conducted as duplicates. Mean values were compared using Student’s *t*-test, and the difference between the results was considered significant when the *p*-value was less than 0.05.

## Figures and Tables

**Figure 1 molecules-27-08239-f001:**
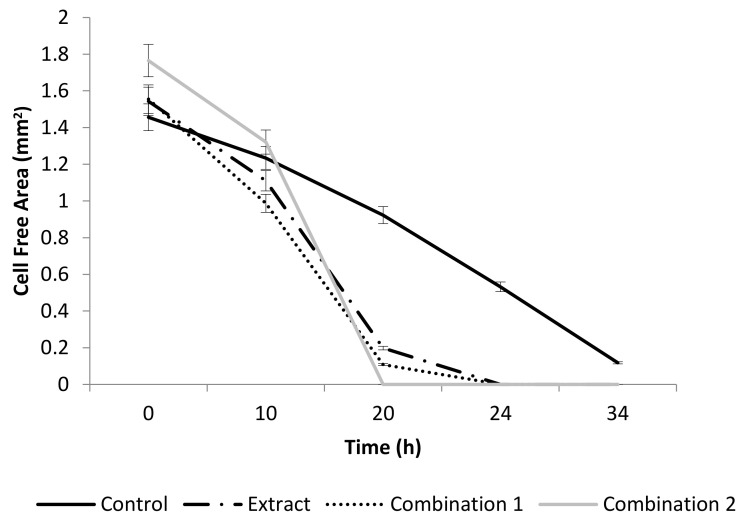
The effect of *A. strigosa* extract and compounds on wound healing in vitro. The rate of gap closure in cultured human dermal fibroblasts (scratch assay: in vitro model of wound healing) at 0, 10, 20, 24, 34 h after wound generation. Control was untreated fibroblasts. Crude extract was added at a concentration of 50 µg/mL. The combination 1 of tested compounds included: quercetin 3-*O*-rutinoside at a concentration of 50 µg/mL, ellagic acid at a concentration of 10 µg/mL and kaempferol 3-*O*-β-rhamnopyranosyl(1→6)-β-glucopyranoside at a concentration of 10 µg/mL. Combination 2 of the tested compounds included: kaempferol at a concentration of 10 µg/mL and ellagic acid at a concentration of 1 µg/mL. Data from three independent experiments are shown (mean ± SD).

**Figure 2 molecules-27-08239-f002:**
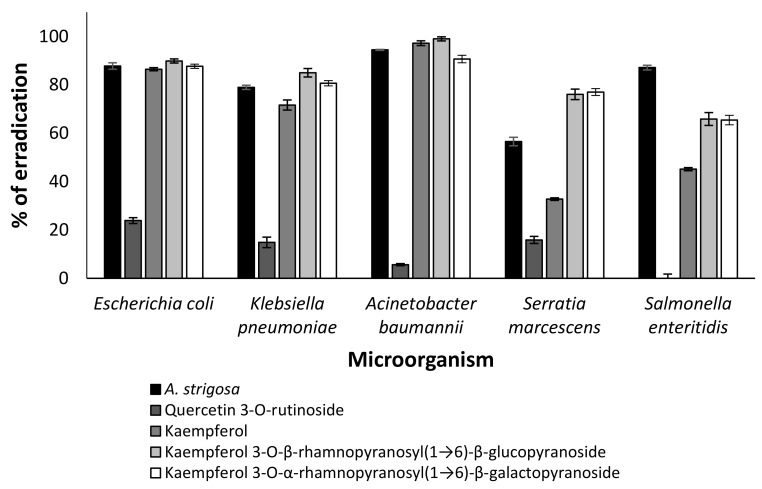
Effect of *A. strigosa* extract and its phytochemicals on eradication of drug-resistant microorganisms (*Escherichia coli*, *Klebsiella pneumoniae*, *Acinetobacter baumannii*, *Serratia marcescens* and *Salmonella enteritidis*). The phytochemicals were applied at concentration of 2 µM. Data from three independent experiments are shown.

**Figure 3 molecules-27-08239-f003:**
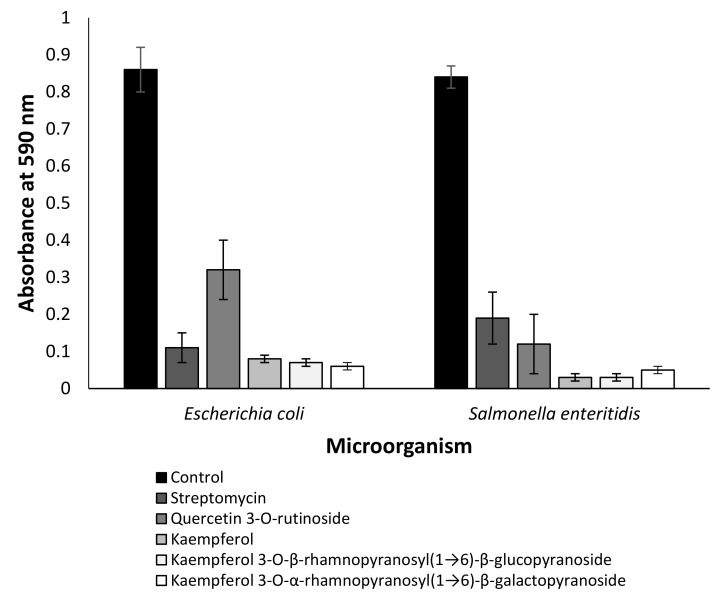
Effect of quercitin 3-*O*-rutinoside, kaempferol and kaempferol glycoside derivatives on biofilm formation. The results are presented as the mean ± SE of the absorbance at 590 nm. The identified compounds and streptomycin were used at a concentration of 2 µM. Data from three independent experiments are shown.

**Table 1 molecules-27-08239-t001:** Identification of phytochemicals of *A. strigosa*.

Compound	Analytical Methods	Concentration, mg/kg	Probability of Compound Identification (%)
Isovaleraldehyde	GC/MS	11.9	87.8
Cubebene	GC/MS	0.9	89.6
2-methylfuran	GC/MS	1.2	93.6
3-methylbutanal	GC/MS	11.9	90.5
Oxirane	GC/MS	2.4	89.7
Octanal	GC/MS	1.6	92.1
Quercetin 3-*O*-rutinoside	HPLC, LC-ESI-MS, MALDI-TOF-MS	5.9	98.7
Kaempferol	HPLC, LC-ESI-MS, MALDI-TOF-MS	1.5	93.8
Ellagic acid	HPLC, LC-ESI-MS, MALDI-TOF-MS	1.8	90.8
Kaempferol 3-*O*-β-rhamnopyranosyl(1→6)-β-glucopyranoside	HPLC, LC-ESI-MS, MALDI-TOF-MS	1.1	87.9
Kaempferol 3-*O*-α-rhamnopyranosyl(1→6)-β-galactopyranoside	HPLC, LC-ESI-MS, MALDI-TOF-MS	1.2	91.7
pyrrolizidine alkaloid	HPLC, LC-ESI-MS, MALDI-TOF-MS	2.3	90.1

GC/MS: Gas chromatography/mass spectrometry. HPLC: High-performance liquid chromatography. LC-ESI-MS: Liquid chromatography–electrospray ionization–tandem mass spectrometric. MALDI-TOF-MS: Matrix-assisted laser desorption ionization mass spectrometry.

## Data Availability

Not applicable.

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
