# Peer review of "Medicinal Properties of Anchusa strigosa and Its Active Compounds"

_molecules, 2022, doi:10.3390/molecules27238239_

Round 1

Reviewer 1 Report

The manuscript reviewed describes a spectrum of biologically active metabolites from the Anchusa strigosa plant, which is widespread in a number of countries in the Middle East, as well as a evaluation of their antimicrobial and wound healing properties. Characteristically, many plants as objects of traditional and alternative medicine are often sources of a whole range of secondary metabolites that are used in therapy. At the same time, a frequent event is the predominance of some of them, in particular, those exhibiting well-defined biological properties. This work has a good basis and relevance, however, from a methodological and productive point of view, it contains a number of shortcomings:

1. Table 1 - the wide range of analytical methods used by the authors of the work to identify compounds from plant materials is puzzling. This is especially true for the methods of liquid chromatography-mass spectrometry and MALDI, the latter method is clearly redundant, because it is more appropriate for polypeptides as well. It is also surprising that the authors declare the probability of identifying these small molecules, which immediately raises questions about. the accuracy of the methods used to measure the masses of molecular ions. In the appendix, it is necessary to provide supporting experimental materials.

2. Page 3 - What is the reason for choosing the effective concentration of 900 µg/ml for the plant extract to test the wound healing properties? Usually use the range of 2-4 mg/ml and more. By the way, the absence of the desired activity at the level of the identified individual compounds is immediately striking, while the extract was active. What could be the reason for this? Maybe the presence of a number of other molecules in significant amounts?

3. Page 4 - in general, it is necessary to conduct a targeted quantitative analysis of these metabolites. Otherwise, a situation arises in which it becomes unclear how many of them are contained in the original active extract.

4. Why were only Gram-negative species selected for testing antibacterial properties?

Reviewer 2 Report

Dear authors,

The manuscript entitled „Medicinal properties of Anchusa strigosa and its active compounds” Yarmolinsky L. et al. describes the phytochemicals of Anchusa strigosa compounds to estimate pro-wound healing (pro-WH) and antimicrobial activities of its active compounds.

After reading the manuscript, I did not notice any significant errors, nor spelling and stylistic errors, English language and style are also fine. The idea of research is interesting and very good. Conclusions adequate to the conducted research.

 Several changes are recommended, and some clarifications are required.

Please explain: A. strigose line 10, line 61, line 62.

Sometimes % is written with space or without space (line 182, line 215 line, 230).

Please italicize in vitro line 99.

You mention Pseudomonas syringae in the methods section, but it is not present in the results.

What are drug-resistant microorganisms in your study?

Do you mean “specific agar surfaces” in line 197? Please explain.

Round 2

Reviewer 1 Report

Dear Authors,

Thanks for your replies. Would you include chomatograms and mass spectra obtained for all compounds identified? In Supplementary Data. 

Author Response

Supplementary Data - mass spectra obtained for all compounds identified.
